# The Effect of Packaging Methods, Storage Time and the Fortification of Poultry Sausages with Fish Oil and Microencapsulated Fish Oil on Their Rheological and Water-Binding Properties

**DOI:** 10.3390/molecules27165235

**Published:** 2022-08-16

**Authors:** Jerzy Stangierski, Hanna Maria Baranowska, Ryszard Rezler, Krzysztof Kawecki

**Affiliations:** 1Department of Food Quality and Safety Management, Faculty of Food Science and Nutrition, Poznań University of Life Sciences, Wojska Polskiego 31/33, 60-624 Poznań, Poland; 2Department of Physics and Biophysics, Faculty of Food Science and Nutrition, Poznań University of Life Sciences, Wojska Polskiego 31/33, 60-624 Poznań, Poland

**Keywords:** poultry sausages, fish oil, microcapsules, vacuum packing, modified atmosphere packing, rheology, low-field NMR

## Abstract

The aim of the study was to investigate how liquid fish oil and microencapsulated oil additives influenced the rheological characteristics and the dynamics of water binding in vacuum-packed (VP) and modified-atmosphere-packed (MAP) poultry sausages during 21-day storage. In contrast to the control sample, the sausages enriched with microencapsulated fish oil (MC) were characterised by the greatest ability to accumulate deformation energy. The elastic properties of all sausage variants increased significantly in the subsequent storage periods, whereas the dynamic viscosity of the samples tended to decrease. This phenomenon was confirmed by the gradual reduction of water activity (A_w_) in all sausages in the subsequent storage periods. The packaging method influenced the dynamics of water binding in an oil-additive-form-dependent manner. During the storage of the VP and MAP sausages, in samples with the fish oil additive the *T*_1_ value tended to increase while the A_w_ decreased. The *T*_1_ value in the MAP MC sample was similar. The FO additive resulted in greater mobility of both proton fractions in the MAP samples than in the VP samples. There were inverse relationships observed in the MC samples. The NMR tests showed that the VP samples with the MC additive were slightly better quality than the other samples.

## 1. Introduction

Poultry meat is a very valuable component of the human diet. It has a high protein content (23–25% in the pectoral muscle), a favourable and fairly stable composition of amino acids (asparagine, arginine, alanine, glutamine, leucine, lysine), vitamins (especially B vitamins), and minerals (zinc, iron) [1]. However, in recent years, traditional meat products, including poultry products, have become the object of innovative meat processing. This trend has resulted from consumers’ concerns about the high content of saturated fats and cholesterol in processed meat products, as they cause heart disease, cancer, and obesity [2,3]. It is noteworthy that for several years cardiovascular diseases (CVD) have been the main cause of deaths around the world [4].

In order to produce typical sausages with a good texture, taste, and aroma, it is necessary to use raw meat containing about 20% fat. Both hard and soft fats can be used, e.g., pork fat, beef fat, mutton fat, poultry fat, and vegetable oils [5]. In recent years there has been greater interest in poultry meat rather than red meat because beef and pork contain more fat. Chicken meat is traditionally leaner than beef or pork, and it does not contain so much cholesterol. In order to prevent health risks and promote healthier consumption of processed meats, the meat industry is taking steps to change the source of fat in products [6]. Fish oil is an interesting alternative. It is used for product fortification or as an animal fat substitute. It is a rich source of polyunsaturated fatty acids (PUFA), especially long-chain n-3 fatty acids, which have health-promoting functions, because they support the cardiovascular, neurological, and immune systems [7]. However, it is not recommended to add fish oil directly to meat batter because fatty acids are highly susceptible to oxidation. This may also cause unfavourable sensory changes and lower the functional characteristics and the durability of final products [8,9].

Research has shown that poultry meat can be successfully used in products enriched with n-3 polyunsaturated fatty acids [6,10,11]. It is a challenge to obtain meat products with the polyunsaturated-fatty-acids-rich fish oil additive, which will provide consumers with valuable nutrients and health-beneficial components while maintaining the sensory and storage quality of these products. This goal can be achieved by adding various types of antioxidants or other additives/spices masking the fishy taste and smell or by using microencapsulated oil [12]. Encapsulation is the process of enclosing active ingredients in a matrix of solid particles [13]. The encapsulation process prevents oxidation and evaporation, enables control of the mechanism of release of active compounds in the gastrointestinal tract and has various other beneficial functions, which improve the shelf life and consumer acceptability of finished products [14,15].

When the ingredients of the meat batter are modified in this way, it is necessary to use appropriate packaging methods to ensure the quality and safety of the finished product. Vacuum packing (VP) and modified atmosphere packing (MAP) are the two most popular methods of packing meat and meat products. The removal of oxygen from the package not only limits the growth of microorganisms, prevents food oxidation, changes in colour, taste, and smell, but it also guarantees the nutritional properties of products, extends their shelf life, and improves the conditions of the presentation of products in refrigerated display cases [16]. Modified atmosphere packing usually consists in replacing the air with a mixture of gases. The process is an extension of the earlier vacuum packing system. Cold cuts are usually packaged in an atmosphere with various proportions of N_2_, CO_2_, and O_2_ [17].

The results of our research are in line with the abovementioned trends. This manuscript describes the experiment conducted on industrially produced, finely comminuted poultry sausages. The aim of the study was to investigate how the liquid and microencapsulated fish oil additives influenced the rheological properties and the dynamics of water binding in poultry sausages depending on the packaging method (VP and MAP) and storage time. The rheological research method enabled the identification of changes occurring in the protein–water–fat dispersion system during its storage. In contrast to the texture analysis, the dynamic mechanical analysis enabled the assessment of the influence of the variable composition of poultry sausages and their storage on the rheological properties and the functional characteristics of the products at the molecular level. The analysis of relaxation times during the storage of model products enabled the observation of changes in the molecular dynamics of water. This analysis led to the conclusions about the quality and durability of the products and it significantly broadened the knowledge acquired in other experiments on the functional characteristics of these products.

## 2. Results and Discussion

### 2.1. Basic Composition

The basic composition of the poultry sausages is shown in Table 1. There were no statistically significant differences between the control sample and the samples with both fish oil additives, i.e., FO and MC (*p* > 0.05). There was no significant effect observed because the meat batter contained small amounts of fish oil and microcapsules, i.e., 7.1 and 11.9 g/kg, respectively. There were similar results from the studies conducted by other researchers. Jiménez-Martín et al. [18] conducted a study on chicken nuggets, Kawecki et al. [11,19] examined poultry sausages, whereas Solomando et al. [20] conducted an experiment on cooked and dry-cured sausages. None of the researchers observed a significant effect of the microencapsulated fish oil additive on the composition of poultry meat products.

However, the sample containing fish oil had a slightly lower content of water (67.4%) and protein (18.2%), but a slightly higher fat content (11.6%) than the control sample. The sample with the microencapsulated fish oil additive had the lowest water content (67.2%) but the highest protein content (18.6%). There was a slightly lower water content observed in studies conducted on pork salchichón as well as raw and cooked pork burgers enriched with fish oil microcapsules [21] and in a study on poultry sausages [10]. The same dependence was also observed by Aquilani et al. [22] in Cinta Senese burgers fortified with omega-3 fatty acids. According to the researchers, this effect was caused by the addition of extra dry matter due to the incorporated microencapsulation material. Josquin et al. [23] observed a similar effect—the moisture content in the sausages with encapsulated oil was about 10% lower than in those with pure fish oil. This may have affected the final protein content in the sample.

### 2.2. pH Analysis

The pH value influences various phenomena and processes, such as protein properties, denaturation, gelling, enzyme activity, the development of microorganisms, and chemical reactions. When pH changes, so does the solubility of proteins and the viscosity of the system, which moves them away from the isoelectric point. This is the result of repulsion between mono-charged groups and interactions between hydrophilic groups and water dipoles. As a result, the hydrophobic interactions between proteins and lipids decrease and so does their ability to emulsify fats. In consequence, the concentration of the protein responsible for the formation of matrices maintaining the water–fat emulsion decreases, which affects the rheological properties and texture of finished products [24]. Therefore, the knowledge of the influence of pH and its control during processing and storage is necessary to make safe products of a high quality and added value.

Our study showed that in subsequent storage periods there were changes in the pH value of the samples, which also depended on the sausage packaging method (Figure 1). The initial pH values measured on the first and seventh days of storage were similar in all sausage variants regardless of the packaging method—they ranged from 5.73 to 5.80 (Figure 1). These values were slightly lower than the values noted in other studies on similar samples of VP and MAP sausages. However, the pH value in individual samples of sausages changed in a similar manner [11]. On the first date of measurements, the samples of sausages with the liquid fish oil additive had the lowest pH (5.73 on average). However, in the next three test periods, the VP sample had the highest pH. A similar dependence was also observed in the MAP samples. In general, in the last two storage periods the samples of sausages with the liquid fish oil (FO) had significantly higher pH values than the other samples (*p* > 0.05). Other authors also observed a similar trend in their studies [6,11,20]. According to these researchers, this increase may have been caused by the protein component and the lipid material in the oil-in-water emulsion. However, Muguerza et al. [25] observed a reverse trend—the control sample and the sample with the microcapsules were characterised by the smallest pH difference, regardless of the packaging method. In the last two periods of the study (day 14 and day 21) the samples with the microcapsules had the lowest pH.

The analysis of the last two periods of the study showed that the MAP samples had lower pH values than the VP samples. It may have been caused by the presence of CO_2_ in the gas mixture. Carbon dioxide dissolves during storage, which may result in a lower pH. Earlier studies showed that during the 21-day storage of poultry sausages the CO_2_ content decreased from about 28% to 20%. At the same time, a decrease in the pH value of stored sausages may also have been caused by increased production of lactic acid resulting from the metabolism of lactic acid bacteria in the product [11,26].

### 2.3. Water Activity

The quality of food products is determined not only by the amount of water but also by the thermodynamic nature of its interactions with other ingredients. The type and the scale of these reactions depend both on the molecular structure of individual components of the system and their different affinities for water (hydrophobic or hydrophilic nature) [27].

Figure 2 shows the equilibrium water activity (A_w_) in the poultry sausages. The mean initial A_w_ values (day 1) of the sausages were similar and amounted to 0.860 ± 0.002. Neither the form of the oil additive nor the packaging method had significant influence on the value of this index (*p* > 0.05). Other researchers made similar observations in their studies on poultry sausages [11]. The packaging method did not have significant effect on the Spanish chorizo sausage [28] and salchichón packaged under different gas conditions [17].

In the subsequent storage periods (days 7–21), the Aw value decreased gradually in both samples, regardless of the packaging method. On the 14th and 21st days of storage, the A_w_ value in the VP samples with the liquid oil additive was lower than in the MAP samples with the microcapsules. It is likely that the hydrophobic properties of the oil added to the batter caused a slight displacement of water from the system. At the same time, the collagen shell of the microcapsule may have caused a slight uptake of water from the system and contributed to a higher A_w_ value in these samples.

In general, the A_w_ value in the VP sausages was slightly lower than in the MAP samples. The vacuum packaging method may result in a slight leakage from sausages. Our observation is consistent with the results of the study conducted by Stasiewicz et al. [29]. A greater leakage from vacuum-packed meat products is the result of negative pressure exerted on the product inside [30]. The loss of juice from meat may affect the measurement of water activity and the rheo-mechanical and sensory analysis of model products.

The PCA method allows us to present the relationships between the samples in one graph. Figure 3 shows the results of principal component analysis (PCA) based on correlations applied to the three types of samples (CO-1, FO-2, MC-3), type of packing (VP-1, MAP-2), and stored for three weeks. The presented figure shows the relationship between the dependent variables (pH, A_w_, *G*′) and independent variables (type of sample, storage time, packing methods). PC 1 (40.63%) and PC 2 (25.02%) accounted for 65.65% of the total variance. There was no correlation between the type of samples, A_w_, and pH. However, there was a positive correlation between type of sample and the modulus elasticity (*G*′). Moreover, the analysis showed a weak positive correlation between packing methods and the modulus elasticity (*G*′). Additionally, the analysis showed a strong negative correlation between the time of storage and water activity.

### 2.4. Rheological Properties

The properties of muscle proteins causing interaction in the water–protein system (water-binding capacity), the association of proteins with fat (fat-emulsification capacity), and protein aggregation (gelling capacity) play a significant role in the production of medium- and finely-comminuted meat products [31]. From the physical point of view, meat batter is a dispersion system consisting of two phases. The hydrocolloidal continuous phase includes an aqueous colloidal solution of muscle proteins and the connective tissue as well as emulsions formed from fat and water-soluble or saline-soluble proteins. The dispersed phase consists of insoluble elements of the muscle and adipose tissues [10,31]. The final product is the result of the thermal fixation of the formed system [31]. The stability of such a system depends on the properties of muscle proteins, the content of water, meat, fat, salt, the addition of non-meat proteins, as well as mechanical and thermal factors [5].

As results from the research conducted so far, the values characterising the rheo-mechanical properties of finely-comminuted meat products reflect the spatial reaction of the protein matrix to the mechanical impact and the physical condition of the meat emulsion constituting the continuous phase of the system. To a lesser extent, they reflect the structural parameters of the dispersed phase consisting of fragments of the muscle tissue [10,32]. The type of fat also significantly affects the rheological properties of meat products as it is one of the main components of the continuous phase of meat batter. The type of fat influences the degree of saturation of fatty acids and the size and distribution of fat globules in the batter [33].

The rheo-mechanical properties of restructured systems are not only the effect of changes in their molecular structure occurring as a result of the applied heat treatment [10,34,35], but also changes in their physicochemical properties conditioned by the packaging method and storage time [36,37]. This fact was confirmed by measurements of the rheological parameters of the vacuum-packed (VP) and modified-atmosphere-packed (MAP) finely-comminuted poultry sausages with the fish oil and microencapsulated fish oil additives, which were cold-stored for 21 days.

Throughout the entire storage time, both the VP and MAP experimental sausages with the fish oil (FO) and microencapsulated oil (MC) additives tended to accumulate elastic energy (*G*′) more than the control sausage (CO) (*p* < 0.05) (Figure 4). The higher *G*′ values of the experimental sausages (FO) were determined by three factors. First of all, liquid fats disperse more easily and are emulsified much faster than solid (animal) fats. Second of all, a higher fat content in the batter increases the ionic strength in the aqueous phase, which improves protein extractability and thus improves the fat emulsification capacity [38]. Third of all, a higher fat content in the batter contributes to the formation of a more concentrated and consistent continuous phase of the emulsion. As a result, a compact structural system of the spatial dispersion of the batter ingredients is formed. As a result, the binding of fat to the structure of the finished product improves and there is increased water-holding capacity after the heat treatment [39]. The continuous phase of the emulsion only affects the dissipative properties but has no effect on the elastic properties. This is reflected by the highest values of dynamic viscosity *η* (Figure 5) and *tgδ* (Figure 6) (*p* < 0.05). There were similar results from earlier studies [10].

Immediately after the production, the sausages containing fat microcapsules (MC) had the greatest ability to accumulate elastic energy (*G*′) (*p* < 0.05). This trend was observed in all periods of the cold storage of the VC and MAP sausages. The same trend was also observed in earlier studies. The increase in the *G*′ value in similar poultry meat products positively correlated with the increase in texture parameters such as hardness, gumminess, and chewiness. At the same time, despite the changes in the rheological and textural traits, the sensory characteristics of the model samples did not deteriorate [10].

It is most likely that this effect was caused by the share of collagen providing a shell to the oil microcapsules, which translated into an increase in the effective density of the segments of the spatial protein matrix maintaining the emulsion composed of water, fat, and proteins [40]. This also translated into the greater hardness of the poultry meat samples containing microencapsulated fat. Kawecki et al. [11] made similar observations in their study. The increase in the density of the segments of the spatial protein matrix maintaining the emulsion composed of water, fat, and proteins was also reflected by lower values of dynamic viscosity *η* (Figure 5) and mechanical energy dissipation *tgδ* (Figure 6) (*p* < 0.05).

As the storage period extended, the *G*′ value (Figure 4) increased significantly (*p* < 0.05) in all the systems of model sausages, whereas the *tgδ* value (Figure 6) decreased, regardless of the packaging method. This means that the elastic properties of the products increased. This effect may have been caused by the reorganisation of the molecular structure, which resulted in an increase in the degree of cross-linking of the spatial protein matrix. One of the causes of this effect was the partial expansion of the existing lattice nodes by binding new macromolecule segments. Another equally important cause was the respiralling of polypeptide chains due to refrigerated storage [32]. This led to the expansion of the spatial lattice nodes by binding new macromolecular segments, which were formed by cross-links between protein polypeptide chains. This conformation change promoted the association of free water molecules, which were able to bind to previously unavailable hydrophilic groups of polypeptide chains. This phenomenon was confirmed by the gradual decrease in the value of water activity in the subsequent periods of storage of all samples of the sausages under analysis.

As results from our research, the comparison of the experimental sausages with the liquid fish oil and microencapsulated oil additive with the standard sausage both immediately after the production and throughout the storage of these products did not reveal any considerable changes in their rheo-mechanical properties, regardless of the packaging method applied. The differences in the rheo-mechanical properties resulted from the changes in the molecular structure of the sausages which occurred during their storage. They were related to the degree of fat and water binding in the protein systems of the sausages.

It is possible to obtain more information on the behaviour of water molecules in model systems of poultry meat products with modified composition and observe their influence on the final quality of the products by conducting physical analyses which are sufficiently sensitive to detect changes in the intermolecular interactions occurring in these systems. For example, nuclear magnetic resonance, especially low-field nuclear magnetic resonance (LF NMR) relaxation, is a method of identification of interactions between water molecules in a system under study. The technique provides information on the molecular mechanisms in the protein-fat systems while preserving the microstructure of the biopolymer system.

### 2.5. Measurement of Dynamics of Water Molecules with LF NMR

The changes in the relaxation parameters were compared according to the type of sample, packaging method, and storage time. Measurements of the spin-lattice (*T*_1_) and spin-spin (*T*_2_) relaxation times in meat products enable the analysis of changes in the dynamics of water molecules in the system and the quantitative interrelations between the bound and unbound water fractions in the system.

As was mentioned before, the meat batter used in the experiment is a two-phase system. It contains sodium chloride ions, which increase the solubility of myofibrillar proteins and thus increase the overall negative charge of the protein. This increases the unfolding of the proteins and shifts their isoelectric point, thus affecting the water absorption of the system. Such an emulsion contains both water protons and fat protons [41]. The analysis of the molecular structure of this system reveals two water fractions, i.e., the bulk fraction and the bound fraction. The former fraction, which is characteristic of oil-in-water emulsions, is mainly distinguished by water–water interactions. In the latter fraction, water interacts with the protein matrix of the system. It is an example of a water-in-oil (W/O) emulsion. The water contained in the product is mainly in a bound state. It is characterised by very low molecular dynamics due to the binding of water molecules by protein chains and the trapping of these molecules in W/O emulsion systems. However, some of the water particles in such systems are in a state known as the bulk fraction. These are water molecules embedded in the spaces of the biopolymer network, which act as a solvent for globular proteins evacuated beyond hydrophobic areas [42,43].

Thanks to the LF NMR method, it is possible to determine the relaxation times describing the possibility of transferring the energy absorbed by protons to the environment (spin-lattice relaxation time *T*_1_) and to adjacent molecules containing protons (spin-spin relaxation time *T*_2_). The applied frequency of the spectrometer and its technological parameters enable separate analysis of the dynamics of molecules of both bulk and bound water fractions through the values of the spin-spin relaxation time components. The values of the spin-lattice relaxation times are one figure which defines the relation/ratio of the amount of water in the bulk fraction to the amount of water in the bound fraction. An increase in the *T*_1_ value represents an increase in the amount of water in the bulk fraction in relation to the amount of water in the bound fraction. The variation in the value of this parameter shows the relative changes in the ratio of the amount of water in both fractions [44,45].

One relaxation time, *T*_1_, and two components of relaxation time, *T*_2_, were determined in the analysis of the poultry meat products. The analysis of the molecular dynamics of protons revealed significant differences between the samples depending on the packaging method and the form of the oil additive (*p* < 0.05). On the first day, the VP samples were characterised by the smallest differences in the mutual ratios of the content of bound and unbound water (Figure 7). The highest *T*_1_ values in the sausages with the fish oil additive (FO) were measured in the first two periods under analysis. This effect may have been caused by the hydrophobic nature of the oil itself, which resulted in fewer water molecules being bound to proteins. This explanation was also confirmed by the decrease in the values of *T*_2_ components in relation to the control sample. An inverse relationship was observed in the MC sample; it may have been caused by the slightly higher protein content in it. A very similar tendency was also observed in the study on poultry batter conducted by Kawecki et al. [45].

In the last two storage periods (days 14 and 21), the lowest *T*_1_ values were observed in the VP sausage sample with oil microcapsules. Kawecki et al. [45] also observed the lowest values of relaxation time *T*_1_ in the poultry sausage batter with the microencapsulated fish oil additive. There were significant changes in the MAP samples. The FO and MC samples had lower *T*_1_ values than the CO sample (day 1). In the CO samples analysed after the first day, both proton fractions were characterised by similar molecular dynamics regardless of the packaging method (*T*_21_ and *T*_22_). However, the packaging method influenced the dynamics of both proton fractions in a manner dependent on the form of the oil additive. The FO additive increased the mobility of both proton fractions in the MAP samples, which was greater than in the VP samples. There were inverse relationships observed in the MC samples.

The changes which occurred in the molecular parameters of the tested systems during storage showed that the oil additive stabilised the ratio of the bound and bulk water fractions to one another in the VP samples (Figure 8). By contrast, when the MAP method was used, on the 14th day of storage the amount of water in the bulk fraction increased and was greater than in the bound fraction. At the same time, the value of *T*_22_ in the bulk fraction decreased significantly (several times) (Figure 9). This suggests that the MAP method significantly changed the dynamics of molecules in the bulk water fraction. It is likely that the liquid oil additive caused the formation of a water-in-oil emulsion as the value of *T*_21_ decreased over time. Regardless of the packaging method, both forms of the oil additive stabilised the mobility of the molecules in the bound water fraction for up to 14 days. Later, the mobility decreased due to the change of the state from bound to bulk or even due to the evacuation of water from the system.

The research showed that the storage of the CO sausages packed in a modified atmosphere resulted in the least significant changes to the *T*_1_ values (Figure 7). In the subsequent research periods, this sample was characterised by lower values than the initial value. The storage of the VP samples resulted in a drastic increase in the *T*_1_ value, which should be interpreted as a significant increase in the amount of unbound water. Regardless of the packaging method, the bound water in the CO samples monotonically reduced the value of the short component of the spin-spin relaxation time *T*_2_. At the same time, the VP sausages were characterised by lower mobility of water molecules than the MAP samples. Throughout the storage time, the VP samples with the fish oil additive (FO) were characterised by a lower amount of the bulk water fraction in relation to the amount of the bound water fraction. The research showed that in comparison with the MAP method, the vacuum packaging of the sausages with the liquid oil additive significantly increased the mobility of protons in the bulk fraction during storage. It is likely that this effect was caused by the formation of emulsion. From the first day throughout the entire storage period, the *T*_1_ value in the MAP sausages with the FO additive was increasing continuously. This may suggest lower microbiological safety of these products due to the increase in the amount of water available to microorganisms. There were not such significant changes in the VP samples. Kawecki et al. [11] conducted microbiological tests on poultry sausages with the refined oil additive and microencapsulated fish oil after 21 days of storage. The tests showed that neither the recommended limit of the total count of colony-forming units nor the count of lactic acid bacteria were exceeded in the sausage samples.

## 3. Materials and Methods

### 3.1. Sausage Production

The experiment was conducted on industrially produced poultry sausages, which were mostly composed of chicken breasts—60%, chicken legs—22%, and chicken skins—18%. A standard set of functional additives was used in the production process, i.e., salt—1.4%, sodium ascorbate—0.1%, spices and aromas—3.2%, and water—20%. The raw meat was comminuted in a grinder through a sieve with a mesh diameter of 5 mm. The other ingredients were comminuted in a cutter. All the other ingredients were added to the comminuted meat mass and then mixed evenly for 5 minutes. The resulting batter was divided into three parts: a control sample (CO), a sample with fish oil added (O), and a sample with fish oil microcapsules added (ME). An appropriate amount of a liquid fish oil preparation or microencapsulated oil was the last ingredient added to two portions of the batter. The final temperature of the batter was 8–10 °C. The batters were stuffed into cellophane casings with a diameter of 13 mm. The next step of the production process involved standard technological treatments, i.e., deposition, dyeing, smoking (60 °C), steaming (75 °C), drying, and cooling. Next, the sausage samples were vacuum-packed (VP) or modified-atmosphere-packed (MAP) with 5 pieces in a package and stored at a temperature of 4 °C ± 2 °C.

The following two oil preparations from © DSM Nutritional Products Ltd. were used in the study:-MEG-3 TM 30% 8a food oil consisting of refined fish oils;-MEG-3 TM 30% powder fish oil microencapsulated in pork gelatine.

According to the manufacturer’s certificate, the total content of eicosapentaenoic acid (EPA) and docosahexaenoic acid (DHA) in the MEG-3™ 30% 8a food oil preparation was 25%, whereas the total content of EPA and DHA in the MEG-3™ 30% preparation was 15%. The inclusion of EPA and DHA in the oil and the powder formulation resulted in a fatty acid content of 2.0 g/kg in the finished products. The aim was to achieve an EPA + DHA content of not less than 80 mg per 100 g and per 100 kcal of the finished product.

The same ingredients, formulation, and technology were used in the production process of three batches (CO, O, and ME) in two different weeks (two replicates).

Each time before the tests, new packages of sausages were taken from the refrigerator and left for about 2 h to reach room temperature. Then, the packages were opened and samples were collected for tests.

### 3.2. Basic Chemical Composition

The following standards were applied to determine the chemical composition of poultry sausages: moisture content [46], protein content [47] with 6.25 as a conversion factor, fat content [48], and ash content [49]. The composition was analysed during the first period of storage for the sausages.

### 3.3. pH Measurement

The pH value was measured with a portable digital HI 99, 161 pH-meter (Hanna Instruments, Eibar, Spain) equipped with an FC2023 glass electrode. The electrode was calibrated with a pH 7.0 buffer (Merc, Germany).

### 3.4. Water Activity Measurement

Water activity (A_w_) was measured with a HygroPalm 23-AW (Rotronic Instruments UK Ltd., Crawley, UK). Before each measurement, the chamber was dried to A_w_ = 0.06.

### 3.5. Rheological Properties

The rheological properties were measured with a dynamic mechanical thermal analyser (DMTA) (COBRABiD—Poznań, Poland). A parallel plate with a 50 mm diameter and 1.5 mm gap measuring system was used. The sample perimeter was covered with a thin layer of high-temperature-resistant silicone grease (GE Electronics, Rockford, IL, USA) to prevent dehydration of the sample edge and moisture evaporation from the sample. The following components of the complex modulus of elasticity were calculated: modulus of elasticity (*G*′), loss tangent (*tgδ*), and dynamic viscosity (*η*). The *G*′ is associated with the part of potential deformation energy which is maintained in the course of periodical deformations. The *tgδ* is a measure of internal friction. It describes the relative quantity of energy dissipated in the material in the course of one deformation cycle. The frequency of own vibrations of the systems amounted to 1.2 Hz. The measurements were taken at 22 °C. The temperature of the chamber and measurement plate were measured with an accuracy of ±0.2 °C. The linear viscoelastic region of each sample was taken into account. The rheological properties of the samples were measured in six replicates.

### 3.6. LF NMR Measurements

Each time, the stored samples of sausages were prepared for LF NMR tests just before the measurement. A cylinder (diameter–0.8 mm, height—10.0 mm) was cut from the sausages, placed in a glass graduated test tube, and sealed. Three samples were collected from each sausage type, always from the centre of the bar.

Spin-lattice *T*_1_ and spin-spin *T*_2_ relaxation times were measured with a pulsed NMR spectrometer operating at a frequency of 15 MHz. Standard inversion-recovery (*T*_1_) and CPMG pulse train (*T*_2_) sequences were used. The 90° pulse length was set at 2.2 µs. The distances between the pulses in the inversion-recovery sequence were varied from 10 to 500 ms to generate 32 different pulse trains. For measurements with the CPMG pulse train, the distance between the pulses ranged from 0.7 to 0.9 ms (depending on the sample). A sequence of 100 spin echoes was applied with 5 signal accumulations. A repetition time of 15 s was used for both sequences. The measurements were made at a controlled temperature of about +20 °C.

The relaxation times were calculated by adjusting the magnetisation regrowth amplitudes *M*(*t*) and the spin echo amplitudes *A*(*t*) to the following formulas:(1)$$M\left(t\right)=M_{0}\left\{1-2e^{\frac{-t}{T_{1}}}\right\}$$
(2)$$A\left(t\right)=A_{0}\overset{2}{\underset{i=1}{\sum }}p_{i}e^{\frac{-t}{T_{2i}}}$$
where:M_0_—the equilibrium amplitude of longitudinal magnetisation,*A*_0_—the equilibrium amplitude of the spin echo,*t*—the distance between the pulses in the respective measurement sequences.

The measurement results showed a monotonic magnetisation regrowth *M*(*t*) and a bi-exponential decay of the spin echo amplitudes *A*(*t*). Therefore, there are two fractions of protons relaxing with two times *T*_21_ and *T*_22_ in Formula (2).

### 3.7. Statistical Analysis

The basic chemical composition, pH, and water activity measurements were analysed statistically with the SPSS software ver. 13.0 (SPSS Inc., Chicago, IL, USA). The results were expressed as mean ± SD. Tukey’s post hoc test was used for the comparison of group means. One-way ANOVA was used to test the significance (LSD test) of the results at *p* < 0.05. Principal component analysis (PCA) was applied as the first step of data analysis to visualize information and detect patterns in the data.

## 4. Conclusions

The research showed that both immediately after the production and throughout the storage of the experimental sausages with the liquid and microencapsulated fish oil additive the rheo-mechanical properties of the experimental products were not worse than those of the standard sausages. The values of the elastic properties of the sausages increased whereas the viscosity of the sausage systems decreased in the subsequent research periods. The differences in the rheo-mechanical properties of the sausages resulted from changes in their molecular structure during storage. These changes were related to the degree of fat and water binding in the protein systems of the sausages. They were also conditioned by the packaging method, which affected the qualitative and quantitative molecular parameters of water in the system. The analysis of the molecular properties of the experimental sausages showed that the MC additive could be used as an additional carrier of health-beneficial ingredients. However, vacuum packaging is recommended because this method ensures more stable water binding parameters, especially when products are stored for several weeks.

## Figures and Tables

**Figure 1 molecules-27-05235-f001:**
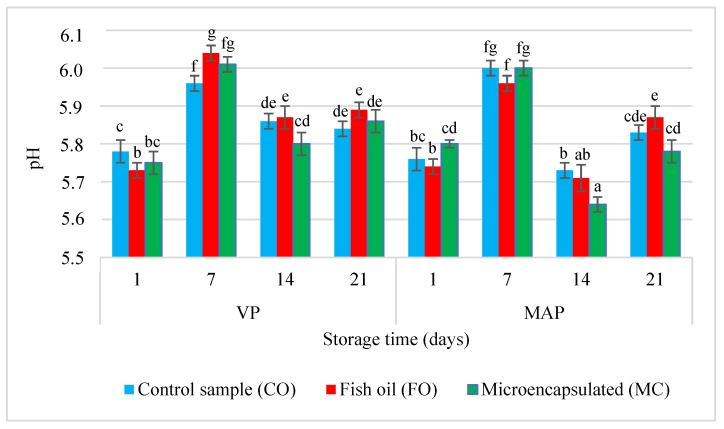
Changes in the pH value of the poultry sausages depending on the type of sample, packing methods, and storage time. a–g = The mean values with the same superscript are not significantly different (*p* < 0.05; mean ± SD; *n* = 4).

**Figure 2 molecules-27-05235-f002:**
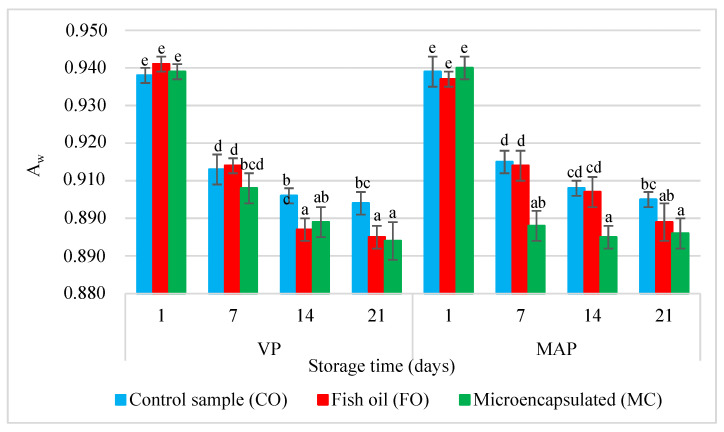
Changes in the water activity of the poultry sausages depending on the type of sample, packing methods, and storage time. a–e = The mean values with the same superscript are not significantly different (*p* < 0.05; mean ± SD; *n* = 4).

**Figure 3 molecules-27-05235-f003:**
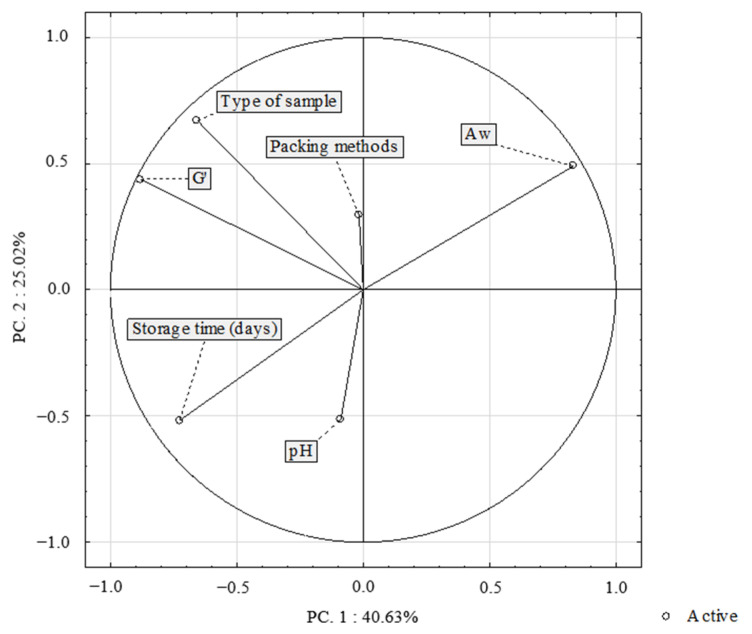
The principal component and load analysis in the analysis of water activity (A_w_), pH, and modulus elasticity (*G*′) depending on the type of sample, packing, and storage time (*n* = 152).

**Figure 4 molecules-27-05235-f004:**
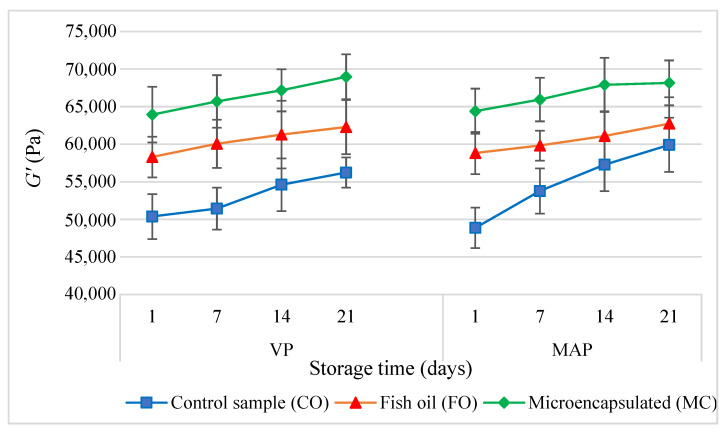
The dependencies of the modulus of elasticity (*G*′) in the poultry sausages during storage (*p* < 0.05; mean ± SD; *n* = 6).

**Figure 5 molecules-27-05235-f005:**
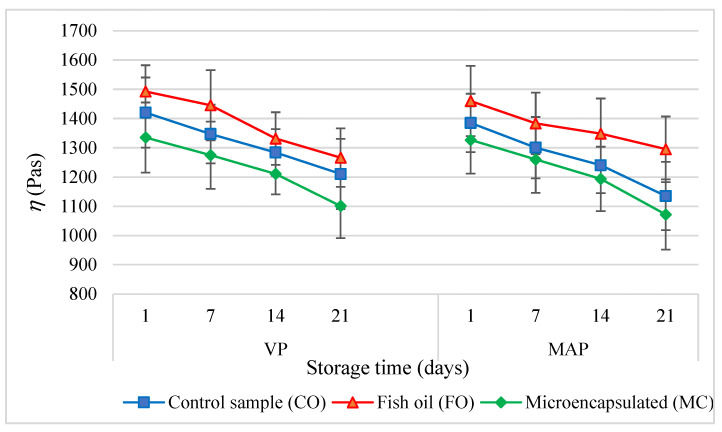
The dependencies of the dynamic viscosity (*η*) in the poultry sausages during storage (*p* < 0.05; mean ± SD; *n* = 6).

**Figure 6 molecules-27-05235-f006:**
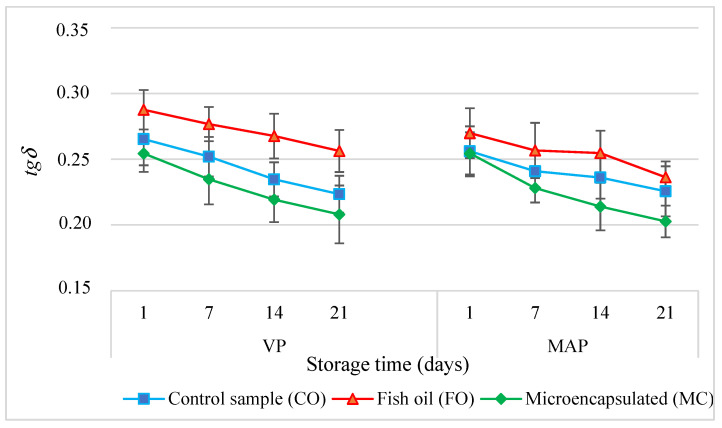
The dependencies of the loss tangent (*tgδ*) in the poultry sausages during storage (*p* < 0.05; mean ± SD; *n* = 6).

**Figure 7 molecules-27-05235-f007:**
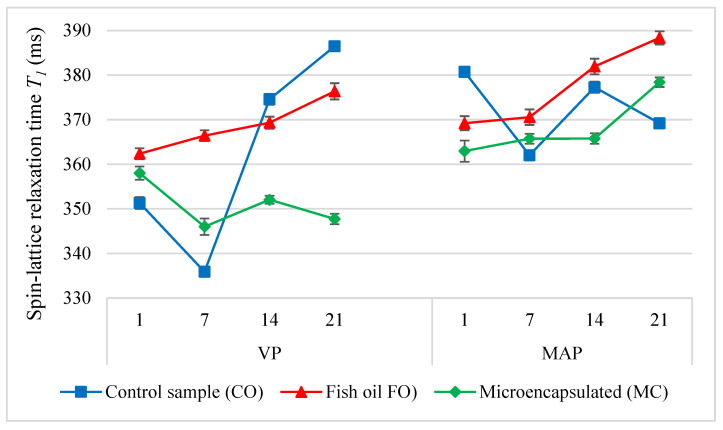
The values of the spin-lattice relaxation time (*T*_1_) in the poultry sausages depending on the packing methods and storage time (*p* < 0.05; mean ± SD; *n* = 3).

**Figure 8 molecules-27-05235-f008:**
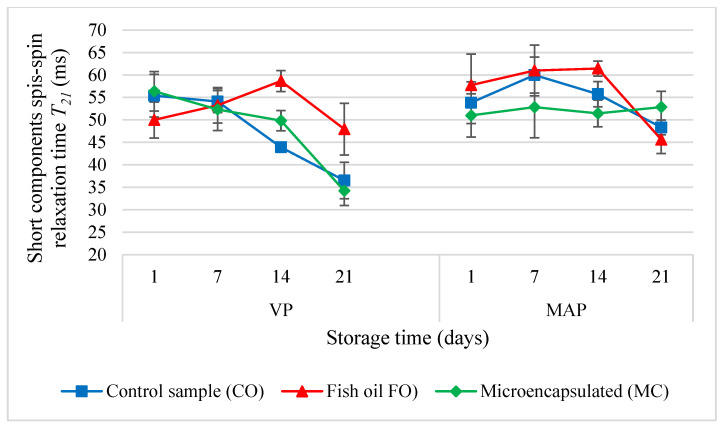
The values of the short components spin-spin relaxation time (*T*_21_) in the poultry sausages depending on the packing methods and storage time (*p* < 0.05; mean ± SD; *n* = 3).

**Figure 9 molecules-27-05235-f009:**
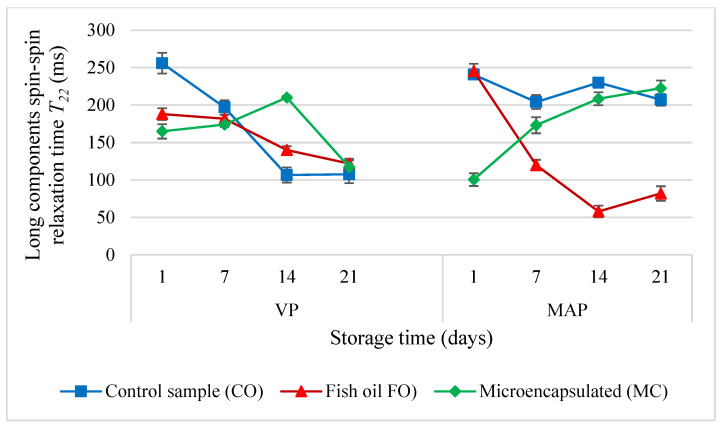
The values of the long components spin-spin relaxation time (*T*_22_) in the poultry sausages depending on the packing methods and storage time (*p* < 0.05; mean ± SD; *n* = 3).

**Table 1 molecules-27-05235-t001:** The basic chemical composition of the poultry sausages.

Chemical		Type of Sample	
Composition (%)	Control Sample (CO)	Fish Oil (FO)	Microencapsulated (MC)
Moisture	67.9 ± 0.6	67.4 ± 0.7	67.2 ± 0.6
Protein	18.4 ± 0.3	18.2 ± 0.4	18.6 ± 0,4
Fat	11.0 ± 0.5	11.6 ± 0.4	11.3 ± 0.4
Ash	2.3 ± 0.1	2.4 ± 0.02	2.2 ± 0.1

No statistically significant differences between means in the same rows (*p* < 0.05; mean ± SD; *n* = 6).

## Data Availability

All data are contained within the article.

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
