# Peer review of "The Effect of Packaging Methods, Storage Time and the Fortification of Poultry Sausages with Fish Oil and Microencapsulated Fish Oil on Their Rheological and Water-Binding Properties"

_molecules, 2022, doi:10.3390/molecules27165235_

Round 1
Reviewer 1 Report
The manuscript refers to the incorporation of liquid fish oil and microencapsulated oil in poultry sausages and their effect on rheological characteristics and water binding during storage in vacuum and modified-atmosphere-packaging.
In general, the manuscript is clearly, although in my opinion requires a minor revision.
Authors did a great job presenting a reasoning discussion in relation to the variations in the elastic properties during storage and their association to the gradual reduction of water activity in all sausages. Also, they pointed that those sausages stored under modified-atmosphere-packaging presented higher variations. It could be desirable to include at least a correlation test that help to describe the relation within the studied variables; in particular, the gradual reduction of water activity may be related with the reduction in pH, the presence of exudate, the microbial development, and the initial emulsion capacity.
Author Response
Thank you for your positive review. First of all, we would like to thank reviewer for taking the time and making the effort to help us improve the above manuscript. We have supplemented the publication with additional statistical analysis. Principal component analysis (PCA) was applied as the first step of data analysis to visualize information and detect patterns in the data.
Reviewer 2 Report
A manuscript entitled: "The effect of packaging methods, storage time and the fortification of poultry sausages with fish oil and microencapsulated fish oil on their rheological and water-binding properties" presents original contribution of authors. Authors clearly describe background and define scope of the investigation. Methodology is described in an appropriate manner. Some changes in the section Results and Discussion are necessary:
1. Lines 102-115: Table 1: There are no statistically significant differences between CO, FO and MC in moisture, protein, fat and ash content. In opposite to these findings authors commented small differences between tested samples and compared with similar findings of the other researchers.
e.g. "Josquin et al. 110 [22] observed a similar effect – the moisture content in the sausages with encapsulated oil was about 10% lower than in those with pure fish oil. This may have affected the final protein content in the sample."
At research of Josquin et al. difference is about 10% and in this research difference between FO and MC is 0.3%. The result of Josquin et al. cannot compare with this research. Also if there are no significant differences between tested samples, comments from lines 102-112 are not appropriate.
Additionally only basic chemical compositions were observed in the CO, FO and MC. Changes of moisture, protein, fat and ash content were not analyzed during the storage time-21 days. Focus of research is on rheological and water-binding properties.
2. Lines 119-120: "When pH increases, so does the solubility of proteins and the viscosity of the system, which moves them away from the isoelectric point."
Solubility of proteins can also change by pH decrease from the isoelectric point (not just by increase), e.g. on a lower pH, -NH2 groups are protonated (repulsion between mono-charged groups) and can react by dipole interactions with water.... Additionally, denaturation process on lower pH exposes protein hydrophobic regions.....
Author Response
First of all, we would like to thank reviewer for taking the time and making the effort to help us improve the above manuscript. Thank you for valuable and detailed comments, which we have very carefully considered.
- The fish oil additive did not significantly affect the basic composition of any of the enriched products. Demonstrated differences in percentages are not large. These results were expected, because only small amounts of fish oil and microcapsules were added to the poultry stuffing, i.e., 7.1 g kg−1 and 11.9 g kg−1, respectively. Basic chemical composition will mainly depend on oil addition and contents of fat in raw meat. References do not show unequivocally direct influence of oil addition and microcapsules on basic chemical composition on meat products. In ours previously published studies no differences in primary ingredients were shown [11,19]. In other publication [46] the difference in water contents between CO and ME samples was determined.
- The basic composition was analyzed during the first period of refrigerated storage for the sausages. Our unpublished studies did not show significant changes in ingredients after 21 day storage period. As it was pointed out by the reviewer focus of research is on rheological and water-binding properties. These analysis were presented in detail in this article.